# Extracorporeal Cardiopulmonary Resuscitation—A Chance for Survival after Sudden Cardiac Arrest

**DOI:** 10.3390/children10020378

**Published:** 2023-02-15

**Authors:** Maria Damps, Michał Buczyński, Łukasz Wiktor

**Affiliations:** 1Department of Anaesthesiology and Intensive Care, Upper Silesian Child Health Centre, Faculty of Medical Sciences in Katowice, Medical University of Silesia, Medyków 16, 40-752 Katowice, Poland; 2Department of Cardiac and General Pediatric Surgery, Medical University of Warsaw, 02-091 Warsaw, Poland; 3Department of Trauma and Orthopaedic Surgery, Upper Silesian Child Health Centre, 40-752 Katowice, Poland

**Keywords:** extracorporeal membrane oxygenation, children resuscitation, cardiac arrest

## Abstract

Extracorporeal membrane oxygenation (ECMO) is an increasingly popular method for the treatment of patients with life-threatening conditions. The case we have described is characterized by the effectiveness of therapy despite resuscitation lasting more than one hour. A 3.5-year-old girl with a negative medical history was admitted to the Department of Cardiology due to ectopic atrial tachycardia. It was decided to perform electrical cardioversion under intravenous anaesthesia. During the induction of anaesthesia, cardiac arrest with pulseless electrical activity (PEA) occurred. Despite resuscitation, a permanent hemodynamically effective heart rhythm was not achieved. Due to prolonged resuscitation (over one hour) and persistent PEA, it was decided to use veno-arterial extracorporeal membrane oxygenation. After three days of intensive ECMO therapy, hemodynamic stabilization was achieved. The time of implementing ECMO therapy and assessment of the initial clinical status of the patient should be emphasized.

## 1. Introduction

Arrhythmias often occur in children. As the results of long-term heart rate monitoring show, premature beats originating from the atria or ventricles appear in most of the examined children. However, in the vast majority of cases, these are very rare beats that occur asymptomatically and do not cause complications. Significant arrhythmias and conduction disorders appear much less frequently. Life-threatening arrhythmias are often caused by a microscopic defect in the structure of the heart, i.e., a focus that generates abnormal stimuli, or an abnormal connection that conducts stimuli parallel to the cardiac conduction system. Supraventricular tachycardia is a very fast heart rate caused by successive contractions of the atria. Such a rhythm may arise as a result of the presence of a focus operating at this speed or as a result of the circulation of an impulse stimulating the atria in the so-called loop. Tachycardia may have a frequency of 200 and in infants even 300 and more beats per minute. It is felt by the child as palpitations, and sometimes it can cause fainting. Tachycardia lasting several hours leads to the development of heart failure. Supraventricular tachycardia always requires prompt hospital treatment. Children with episodes of supraventricular tachycardia are usually qualified for ablation treatment.

Extracorporeal membrane oxygenation (ECMO) is an increasingly popular method for the treatment of patients with life-threatening conditions. Extracorporeal cardiopulmonary resuscitation (ECPR) is the application of extracorporeal membrane oxygenation (ECMO) in patients where conventional cardiopulmonary resuscitation (CCPR) measures are unsuccessful in achieving a sustained return of spontaneous circulation (ROSC) [1]. The time required to establish ECMO support is highly dependent upon the capabilities of the resuscitation team and patient factors [2]. It is reasonable to consider commencing cannulation after 10–20 min of failed resuscitation efforts [3]. Beyond 20 min of refractory arrest, the probability of ROSC and survival with CCPR is <5% [4]. These data refer to adult patients [1]. We have weak recommendation and very low certainty evidence regarding children. The concept of extracorporeal cardiopulmonary resuscitation appeared in the European Guidelines for Resuscitation 2015 [5]. In line with the current guidelines of 2021, and in line with the ILCOR 2019 COSTR update on the use of eCPR in children, we advise considering eCPR for children with ED or IHCA with a presumed or confirmed reversible cause where conventional ALS does not promptly lead to ROSC [6]. The case we have described is characterized by the effectiveness of therapy despite resuscitation lasting more than one hour. Severe arrhythmias resistant to other forms of therapy are indications for veno-arterial ECMO supporting gas exchange and both respiration and circulation.

## 2. Case Report

A 3.5-year-old girlweighing 14 kg, with a negative medical history was admitted to the Department of Cardiology due to ectopic atrial tachycardi—according to the anamnesis, the child felt worse 3 h before reporting to the hospital. The anamnesis revealed abdominal pain, cold limbs, pale skin, and cyanosis around the lips and on the distal parts of the limbs. On admission, the girl was in a moderately serious condition. Heart sounds—loud; pulse on the peripheral arteries—present, weakly tense; liver—about 1.5 cm from under the right costal arch; a vesicular murmur over the lung fields; no peripheral edema; the child was in logical contact appropriate for age. ECG showed supraventricular tachycardia with narrow QRS, HR 240/min (graphics nr 1), blood pressure 85/40 mmHg, heart rate 200/min, lactate 7,6 mmol/l, pH 7,25, BE −11,2 mmol/l. Echo: in m-mode measurements: LVEDD = 38 mm; LVESD = 26 mm; LVFS 30%, regurgitation MV II. Doppler: MV−0.8 m/s, TV −0.6 m/s, AoV−0.6 m/s, PAV −0.4 m/s. After the metabolic disorders were corrected, the child’s general condition stabilized. 

In the Department of Cardiology, an attempt of pharmacologic cardioversion was made. Adenosine was administered three times (1.5 mg and 3 mg twice) and the patient received amiodarone via infusion (5 mg/kg body weight over 2 h, then 6 mg/h); the total dose of amiodarone was 140 mg (10 mg/kg over 12 h hospitalization in the Department of Cardiology). The next morning, due to the lack of clinical effect and the deterioration of circulatory capacity (increased metabolic acidosis), it was decided to perform electrical cardioversion under intravenous anaesthesia.

During induction of anaesthesia (midazolam 0.13 mg/kg body weight, fentanyl 2 μg/kg body weight, propofol 1.3 mg/kg body weight), cardiac arrest with pulseless electrical activity (PEA) occurred. An immediate resuscitation was taken. Approximately two minutes later, hemodynamically effective supraventricular tachycardia with a blood pressure of 62/40 mmHg was achieved. Because of the patient’s unstable condition, cardioversion was performed three times (10 J, 20 J, 30 J), resulting in the transient lowering of patient’s heart rate and repeated cardiac arrest. Another cardiopulmonary resuscitation lasting about 1 min resulted in a return of heart rate (about 216 bpm) with clinical features of pulmonary oedema. The child was urgently transported to the Intensive Care Unit. A few minutes after admission to the ICU, a repeated episode of cardiac arrest with PEA occurred. Cardiopulmonary resuscitation was resumed. Despite resuscitation, a permanent hemodynamically effective heart rhythm was not achieved. Due to prolonged resuscitation (over one hour) and persistent PEA, it was decided to use veno-arterial extracorporeal membrane oxygenation. Open chest procedure during CPR, Ao asc arterial cannula 14 Fr, right atrium 22 Fr, left atrium venting cannula 12 Fr. Maquet RotaFlow pump 100 mL/kg output. Rhythm recovery after 10 h. Venting left atrium had been crucial; before it, there were signs of overloading left ventricle in echo after sinus rhythm recovery (graphics nr2). After three days of intensive ECMO therapy, hemodynamic stabilization was achieved. The echocardiogram showed normal flow through the heart valves, trace regurgitation of all valves, normal dimensions of the heart chambers, slightly impaired myocardial contractility (EF = 43.7%, FS = 21.1%), and no fluid in the pericardial sac. Cardiovascular pharmacological treatment in the ICU included: Adrenaline 0.02 μg/kg/min, Milrinone 0.7 μg/kg/min, Amiodarone 5 mg/kg/day, and Metoprorol 0.5 mg twice daily. Three days after cardiac arrest, ECMO was removed, and on the seventh day the patient was extubated. On the 13th day after cardiac arrest, the child was conscious, verbally responsive, cardiovascularly and respiratorily stable, without neurological abnormalities. The patient was qualified for ablation; no source of tachycardia was detected during examination. Ectopic left atrial tachycardia was confirmed in the child and a decision was made to introduce invasive arrhythmia treatment (ectopic focus ablation) 2 months after ECMO therapy.

At present, the child is in a stable condition, under the regular control of a cardiologist in the outpatient clinic. To understand the dynamics of the events, we present a timeline (graphics nr 3).

Today the patient is 11 years old and is a healthy girl.

## 3. Discussion

Our case is unusual for several reasons. First, the symptoms of heart failure developed dramatically (within a few hours) in a previously healthy child with a negative medical history. Secondly, the initial serious condition of the child, i.e., the symptoms of circulatory failure, microperfusion disorders, and metabolic acidosis stabilized after the use of pharmacotherapy, which could have slightly dulled the vigilance of the duty doctor. However, after a few hours, arrhythmia in the form of supraventricular tachycardia returned and the child’s condition began to deteriorate. Another event worthy of attention and for drawing conclusions for the future is cardiac arrest in the mechanism of pulseless electrical activity (PEA) during the induction of anesthesia for cardioversion. Perhaps if only ketamine had been used instead of the opioid and propofol with the benzodiazepine, there would have been no sudden cardiac arrest (SCA). Of course, we do not know this, but we would like to emphasize the need for special attention when anesthetizing a hemodynamically unstable child for cardioversion. Another aspect of our case report is related to the ongoing resuscitation activities. The final result of our clinical activities emphasizes the need for effective cardiac massage and effective proper ventilation in the first place. Despite a prolonged resuscitation, we managed to maintain proper circulation and oxygenation of the central nervous system. Finally, the last aspect of our case is the fact of using VA-ECMO (veno-arterial extracorporeal membrane oxygenation). We faced a very difficult dilemma—that is, whether, despite such a long-term resuscitation, the patient could benefit from the method of extracorporeal blood oxygenation and whether our actions would not bear the signs of futile therapy. Our decision turned out to be right; however, ethical aspects are extremely important, as evidenced by the addition of a chapter on ethics to the latest Resuscitation Guidelines. The prevention of futile therapy is now part of making the right clinical decisions. We decided to describe our case primarily for educational purposes, which also fits in with the ideas of the Resuscitation Guidelines. There are no more difficult and stressful decisions in medicine than those that must be made quickly and suddenly during a sudden cardiac arrest. 

Extracorporeal cardiopulmonary resuscitation (ECPR) in the paediatric population was initially used for inpatient cardiac arrest in children with congenital heart diseases, especially after cardiac surgery [7]. In children, it should be considered in cases of cardiac arrest resistant to conventional CPR, in the presence of a potentially reversible cause, and if cardiac arrest occurred in a highly specialized hospital that was experienced in extracorporeal life support (ECLS) [5,6].

The time from sudden cardiac arrest to ECPR is a key element of resuscitation [3,8]. The maximum arrest duration before ECPR becomes futile and has also not been well defined. 

Despite promising individual case reports of extracorporeal resuscitation, there is no clear recommendation. Deep hypothermia is the most frequently reported cause of sudden cardiac arrest and efficacy of ECMO in OHCA [9]. E-CPR should be considered early for children with ED or IHCA and a (presumed) reversible cause when conventional ALS does not promptly lead to ROSC, in a healthcare context where expertise, resources, and sustainable systems are available to rapidly initiate ECLS. However, there is still controversy over the use of ECMO for non-cardiac reasons in children. In a meta-analysis of individual patient data in severe ARDS, 90-day mortality was significantly lowered via ECMO compared with conventional management [10]. The available literature suggests a potential benefit for ECMO use in selected poisoned patients with refractory shock, cardiac arrest, or respiratory failure. Resuscitation in children may raise more emotions than in adults, but we are not authorized to implement useless therapy [11]. Cardiogenic shock with low cardiac output syndrome caused by a number of factors, such as anaphylactic shock or isolated heart injury or drug intoxication, though de facto attributable to primarily non-cardiac causes leading to cardiac arrest in previously healthy patients, may provide indications for ECPR use, with a chance for good prognosis. Given the high resources needed and the fact that outcome is related to time to initiation and the quality of CPR before initiation, the indications for eCPR in OHCA are very limited. The results of ECPR studies conducted in adults cannot be extrapolated to pediatric OHCA, given the difference in causes of cardiac arrest between children and adults, the techniques and equipment applied for ECPR, and the postcardiac arrest care interventions [12]. The largest study we have is a report from the American Heart Association’s Get With The Guidelines–Resuscitation Registry [13]. For children with in-hospital CPR of a ≥10 min duration, E-CPR was associated with improved survival to hospital discharge and survival with favorable neurological outcomes compared with C-CPR. Another large study of pediatric IHCA from the American Heart Association’s reported an inverse relationship between CPR duration and survival after C-CRP and found that survival and survival with favorable neurological outcomes declined linearly with each 15-min epoch of CRP [13]. Survival after cardiopulmonary resuscitation for more than 30 min prior to extracorporeal membrane oxygenation cannulation in the pediatric cardiac cohort was 43.8% in the observational retrospective cohort study by Pilar Anton Martin [14]. Factors associated with mortality included calcium use during resuscitation, longer cardiopulmonary resuscitation, increased chest compression pauses during cannulation, the use of continuous renal replacement therapy, and abnormal pupils during extracorporeal membrane oxygenation support. Extracorporeal life support should be considered as soon as possible, but it should not be a futile therapy. These European Resuscitation Council Ethics guidelines provide evidence-based recommendations for the ethical, routine practice of resuscitation and the end-of-life care of adults and children [15]. Clinical decision-making for the benefit of patients and protecting their best interests has become extremely difficult over the years. The multitude of therapeutic options gives the doctor a wide range of procedures to choose from. In addition, although evidence-based medicine and guidelines are of help, we still do not fully know the best management option for a given patient. More and more often it is also being pointed out that clinicians’ experienced emotions that can and do affect clinical decision-making [16]. Of course, the situation varies depending on the cultural and legal conditions of individual countries, and even on individual medical disciplines. However, patient-centered (autonomous) care is undoubtedly the most popular attitude today. A doctor cannot be guided only by medical reasons, leading to the so-called medicalization of death. The preferred attitude, referred to as dignity-conserving care [17], respects the attitude of autonomy towards the experiences and the situation of a particular patient and includes empathy, compassion, and dialogue (ABCDs of dignity-conserving care: attitudes, behaviors, compassion, and dialogue). According to the majority of the respondents in my study [18], the decision to limit futile therapy would be facilitated by the patient’s declaration of will. Trust in the family can be perceived differently, and we do not always agree to let the family decide for us. Of course, in the case of children, we have a more difficult situation. We should be resistant to attempts at manipulation or pressure from the patient’s family and environment, who do not always understand the patient’s good in the same way as the patient himself/herself. In the described case, we decided to use ECMO on our own, without any pressure or persuasion from parents. We were aware that the girl was initially healthy, with a negative medical history, and our cardiopulmonary resuscitation was performed with due diligence. So, we wanted to use the last possible chance to save the patient. Today we can admit that the final effect exceeded our expectations. We were surprised by the great neurological condition of the child.

## 4. Conclusions

The lack of comorbidities and the patient’s negative medical history are crucial for the success of resuscitation. A proper resuscitation performed according to guidelines allows us to gain valuable time to analyze the possible reversible causes of cardiac arrest as well as to implement rescue therapies. The time of implementing ECMO therapy and the assessment of the initial clinical status of the patient should be emphasized. Extracorporeal life support should be considered as soon as possible. The case we describe is unusual due to the time of resuscitation and the lack of neurological deficits.

## Data Availability

Department of Anaesthesiology and Intensive Care, Upper Silesian Child Health Centre.

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
