# Peer review of "Extracorporeal Cardiopulmonary Resuscitation—A Chance for Survival after Sudden Cardiac Arrest"

_children, 2023, doi:10.3390/children10020378_

Round 1
Reviewer 1 Report
In this case report, the authors address a very current topic regarding the use of ECMO therapy in the pediatric population that has suffered a cardiac arrest, following two important directions: the use of ECMO therapy in children who have suffered a cardiac arrest from a non-cardiac cause with the ultimate goal of to obtain post-resuscitation hemodynamic stability and the use of ECMO in cases with prolonged conventional cardiopulmonary resuscitation over 20 minutes - ERCP, regardless of the cardiac or non-cardiac etiology of the cardiac arrest.
Can the authors specify how long after the start of ECMO therapy a rhythm compatible with ROSC was obtained? From your article it is clear when hemodynamic stability was achieved during ECMO therapy but not when the rhythm compatible with ROCS was resumed.
I think it is important to mention if this aspect is also possible considering that the current guidelines divide the indications of ECMO therapy into ECPR - ECMO flow is instituted during conventional CPR and VA ECMO - patients cannulated after 20 min of sustained return of spontaneous circulation.
I appreciate your efforts as authors in writing this case report and the chosen topic.
Reviewer 2 Report
The authors have presented a case report of eCPR used after arrhythmia and PEA arrest. I have the following questions and comments:
1)the 2015 European guidelines have been mentioned in lines 30 and 68. Add reference for these.
2)How was ECMO cannulation performed? What cannulas and sizes were uses?
3)What was the weight of the patient?
4)What did the echo show after ECMO?
5) References 2,4,5,6,8,9 have not been cited anywhere in the manuscript
Reviewer 3 Report
The authors present a very interesting case. Several statements need to be clarified before publication:
#1 “ectopic atrial tachycardia“: The authors should present the ecg.
#2 Any vital signs (blood pressure, heart rate) and lab testing results (lactate, ph, BE etc.), echo -> cardiogenic shock?
#3 “adenosine was administered three times and the patient received amiodarone in infusion”: Please indicate the doses (absolute and /kg)
#4 „During induction of anaesthesia“: The authors should describe the induction regime precisely.
#5 To understand the dynamic (starting with the admission), the authors should consider presenting a timeline.
#6 “it was decided to use veno-arterial extracorporeal membrane oxygenation.” The authors are advised to give much more details about the procedure (duration and way of implantation, flow after implantation…)
#7 “After three days of intensive ECMO therapy, hemo- 55 dynamic stabilization was achieved.” How the authors recognized this stabilization? Describe the weaning procedure etc.
Author Response
Thank you very much for your suggestions. Please see the new version of our paper.
Round 2
Reviewer 3 Report
The authors must address the following questions:
#1 “ectopic atrial tachycardia“: The authors should present the ecg.
#2 Any vital signs (blood pressure, heart rate) and lab testing results (lactate, ph, BE etc.), echo -> cardiogenic shock? [before implantation]
#3 “adenosine was administered three times and the patient received amiodarone in infusion”: Please indicate the doses (absolute and /kg)
#4 „During induction of anaesthesia“: The authors should describe the induction regime precisely.
#5 To understand the dynamic (starting with the admission), the authors should consider presenting a timeline.
